# Photothermal Infrared Radiometry and Thermoreflectance—Unique Strategy for Thermal Transport Characterization of Nanolayers

**DOI:** 10.3390/nano14211711

**Published:** 2024-10-27

**Authors:** Ankur Chatterjee, Mohanachandran Nair Sindhu Swapna, Ameneh Mikaeeli, Misha Khalid, Dorota Korte, Andreas D. Wieck, Michal Pawlak

**Affiliations:** 1Institute of Physics, Faculty of Physics, Astronomy and Informatics, Nicolaus Copernicus University in Torun, Grudziadzka 5, 87-100 Torun, Poland; connecttoankur94@doktorant.umk.pl (A.C.); ameneh.mikaeeli@doktorant.umk.pl (A.M.); mishakhalid@doktorant.umk.pl (M.K.); 2Chair of Applied Solid-State Physics, Faculty of Physics and Astronomy, Ruhr-University Bochum, Universitaetsstrasse 150, D-44780 Bochum, Germany; andreas.wieck@ruhr-uni-bochum.de; 3Laboratory for Environmental and Life Sciences, University of Nova Gorica, Vipavska 13, 5000 Nova Gorica, Slovenia; swapna.nair@ung.si (M.N.S.S.); dorota.korte@ung.si (D.K.)

**Keywords:** thermoreflectance, thermal anisotropy, photothermal radiometry, frequency domain thermoreflectance (FDTR)

## Abstract

Thermal transport properties for the isotropic and anisotropic characterization of nanolayers have been a significant gap in the research over the last decade. Multiple studies have been close to determining the thermal conductivity, diffusivity, and boundary resistance between the layers. The methods detailed in this work involve non-contact frequency domain pump-probe thermoreflectance (FDTR) and photothermal radiometry (PTR) methods for the ultraprecise determination of in-plane and cross-plane thermal transport properties. The motivation of one of the works is the advantage of the use of amplitude (TR signal) as one of the input parameters along with the phase for the determination of thermal parameters. In this article, we present a unique strategy for measuring the thermal transport parameters of thin films, including cross-plane thermal diffusivity, in-plane thermal conductivity, and thermal boundary resistance as a comprehensively reviewed article. The results obtained for organic and inorganic thin films are presented. Precise ranges for the thermal conductivity can be across confidence intervals for material measurements between 0.5 and 60 W/m-K for multiple nanolayers. The presented strategy is based on frequency-resolved methods, which, in contrast to time-resolved methods, make it possible to measure volumetric-specific heat. It is worth adding that the presented strategy allows for accurate (the signal in both methods depends on cross-plane thermal conductivity and thermal boundary resistance) and precise measurement.

## 1. Introduction

One of the major issues facing modern science is temperature management applications. One of the most interesting applications is, for example, the ‘Thermal Management’ application: thermal switch [1], thermoelectric generator [2], thermal mersister [3], and quantum thermal transistor [4]. In order to select components efficiently, knowledge of the thermal parameters is essential. In order to fully characterize materials, knowledge of all transport parameters such as in-plane thermal conductivity, cross-plane thermal conductivity, diffusivity, and the thermal boundary between the thin layer and substrate is essential. However, the problem of parameter measurement remains [5,6,7]. To the best of our knowledge, there is currently no method/strategy that would allow for such a strategy. In this article, we present such a strategy. 

In recent years, there has been a substantial focus on studying the thermal transport properties, such as conductivity, boundary resistance, and diffusivity, in both bulk and low-dimensional materials. The interest in this phenomenon is particularly remarkable in materials with several layers, where the thermal conductivity components in the plane and perpendicular to the plane show significant anisotropy. Furthermore, this gives rise to situations involving “artificial anisotropy” caused by low dimensionality, as seen in semiconductor superlattice thin films. This serves as the driving force for the research. Several experimental techniques have been devised and effectively utilized to investigate the thermal transport characteristics that are perpendicular to the plane. These strategies frequently depend on diverse approaches, such as electrical or optical methods. On the other hand, studying heat transport within a plane is more difficult because most methods are not very sensitive to this aspect, among other factors. Several experimental methods aim to achieve sensitivity to thermal anisotropy within a plane. Several research groups have shown this sensitivity using various experimental setups, and among them, photothermal and thermoreflectance are widely used for different applications [8,9,10,11,12,13,14,15,16,17,18,19,20,21,22,23]. The 3-Omega method, an electrical technique, has been successfully used to examine materials with thermal anisotropy. This approach involves placing the heater and thermometer at different positions on the sample surface. Nevertheless, these methods are particularly appropriate for samples that have electrical insulation properties. When metallic samples are used, it is essential to electrically isolate the metallic transducer from the sample. This can be difficult since there is a risk of current leakage between the transducer and the sample. In addition, electrical approaches likewise necessitate substantial manufacturing endeavors as a distinct sample must be created for each in-plane direction that is intended to be examined. Alternatively, many contactless techniques, sometimes utilizing time- or frequency domain thermoreflectance, have effectively shown the ability to address thermal anisotropy [5,8,9,16,19] in both the time and frequency domains. While the pump-probe approaches have been successful in estimating the thermal conductivity tensor, some areas can be enhanced, as demonstrated by the innovative approach proposed in this study. For instance, the techniques described in (i) and (ii) are often affected by the geometry of the heat source, specifically its spatial energy distribution, which is caused by the limited ability to adjust the distance between the heat source and the probe. Usually, when dealing with continuous heat sources that are modeled as concentrated Gaussian beams, it is not possible to attain significant offsets because the thermal field decays rapidly in space. Nevertheless, the method outlined in (iii) obviates the necessity for a specific understanding of the heat source’s configuration, akin to the 3-Omega technique [14,16]. However, it necessitates an accurate understanding of the temperature coefficient of reflectance of the surface of the sample (or transducer in the majority of instances). This is because the thermal conductivity calculation requires the absolute temperature to be known. 

## 2. Methods

### 2.1. Photothermal Infrared Radiometry (PTR) Method

Figure 1 shows the experimental set-up with the DPSS laser (λ = 532 nm, 800 mW dc power) with an intensity modulated in the range of 100 Hz–10 MHz using an acoustic–optical modulator. The IR radiation emitted from the sample is collected and collimated using two parabolic mirrors (Au-coated, off-axial). The emitted IR radiation is focused onto the detector (mercury cadmium telluride (MCT)) with a window coating of antireflection Germanium. For spectrally resolved measurements, appropriate filters are used in the front of the detector, which is connected to a lock-in amplifier and to a PC, where the signals corresponding to each modulation frequency are analyzed [24].

### 2.2. Thermoreflectance Set-Up (FDTR)

A complete scheme of the frequency-domain thermoreflectance (BO-FDTR) experimental setup is shown in Figure 2 [24]. The pump laser’s wavelength was adjusted to the value λ_pump_ = 488 nm (CNI Laser), whereas the wavelength of the probe laser was set to λ_probe_ = 638 nm (Cobolt 06-01 Series). The power of the pump laser was sinusoidally intensity-modulated with acoustic–optical modulator (model Crystal Technology, Inc., San Jose, CA, USA, AOM 3080-120) in the frequency range between 100 Hz and 1.5 MHz using a 30 MHz arbitrary waveform generator (Agilent 33522A—250 MSa/s, Agilent Technologies, Inc., Santa Rosa, CA, USA), whereas the probe laser Cobolt 06-MLD (modulated laser diode; λ = 488 nm, variable CW power, Stockholm, Sweden) intensity was approximately constant in time. The spot size for individual beams was directly measured using a 24–215 data ray-scanning slit beam profiler. This was staged for the accurate measurement of the beam diameters of 2 µm to 4 mm. The pump laser power was incrementally raised, while the probe laser power was maintained at a low level to prevent any extra thermal effects (usually in the range of a few milliwatts). The lasers were integrated into the primary optical axis using beam splitters (BS) and directed through a dichroic mirror. The resulting beam was then concentrated onto the sample using the objective lens (Mitutoyo MY10X-803, Kawasaki, Japan). The pump and probe beams had a constant diameter of 2.42 μm at the output of the lasers, and their spatial intensity distribution followed a Gaussian pattern. Upon reflection, the pump and the probe beams took the same path retracing back to a photodiode (PD)—PDA8A2—Si Fixed Gain Detector, Newton, New Jersey, US, 320–1000 nm, 50 MHz BW, 0.5 mm^2^, where their intensity was converted into a voltage signal. The pump beam is blocked using a notch filter. The voltage signals from the probe were measured individually and the pump beam was blocked (separated using inline bandpass filters) as a function of each heating frequency and the offset distance. Some additional phase lag was probably acquired by the post-sample optics. The voltage signal was analyzed by the ratio of out-of-phase components to the in-phase component of thermal response for phase and amplitude, while the magnitude of both components for amplitude. 

The typical experimental set-up for FDTR measurements is presented in Figure 2. An intensity-modulated laser is used to heat the sample, whose wavelength is chosen according to the type of material the transducer is made of to ensure maximum value of the reflectance coefficient and, thus, maximum system sensitivity [24].

In our study, we conducted experiments to investigate the impact of the finite dimensions of the focused pump and probe lasers on the phase lag response (between the pump and probe) and the amplitude of the thermal reaction. This section presents typical thermoreflectance data that demonstrate both the amplitude and phase lag variations based on the spatial offset and pump modulation frequency. The beam profiler was used to measure the 1/e^2^ width of the line-shaped pump, which was approximately 2.42 μm. The spot size remained consistent throughout the duration of the experiment. As the spatial offset increases, the phase lag and amplitude of the spot-shaped pump decrease and eventually reach zero when the distance is closer to the diameter of the pump. Therefore, in practical terms, the probing spot can be seen as being point-like. This simplifies the process of analyzing the data and the associated equations, as demonstrated in the preceding section. In addition, it is worth mentioning that taking into account the limited dimensions of the probe beam does not have any impact on the results obtained for thermal diffusivity. This occurs because it simply leads to a regional calculation of the average delay in the phase and amplitude of the thermal reaction. Due to the non-linear spatial dependence of the phase lag, the resulting *φ*(r) will mostly display a Gaussian profile for estimating errors.

### 2.3. Description of PTR Method

In a general case, the PTR signal is proportional to Ref. [25].
(1)$$∆E\approx T^{3}∆T,$$
where *T* is the temperature and ∆T are the temperature oscillations induced in the sample by laser. Figure 3 shows PTR signal amplitude as a function of temperature.

The PTR signal for modulation frequency, *f*; the sample of thickness, *L*; and infrared absorption coefficient, *β_IR_*, can be written as follows [25]:(2)$$S\left(f,\beta_{IR}\right)\propto \int_{0}^{L}\beta_{IR}\cdot ∆T\left(z,f\right)\cdot exp\left(-\beta_{IR}\cdot z\right)dz$$

### 2.4. Description of TR Method

Thermoreflectance is the technique that detects variations in temperature near the surface by relying on temperature-dependent optical reflectance [24]. It is based on the modulation of the sample’s temperature with a metallic layer (transducer) deposited on its surface and monitors the changes in the layer’s reflectivity. When the metal layer’s temperature is increased, the number of phonons inside this layer is also increased, which in turn increases the number of electron–phonon collisions. As a result, the light absorption coefficient decreases, leading to an increase in the light reflection.

The *TR* signal is calculated according to Equation (3) [24]:(3)$$S_{TR}=\frac{∆R}{R}=\left(\frac{1}{R}\frac{\partial R}{\partial T}\right)∆T=C_{TR}∆T$$
where ∆*R*/*R*—relative change in reflectivity, Δ*T*—temperature oscillations, and *C_TR_*—TR coefficient. One of the more important advantages of thermoreflectance over the methods discussed here is the ability to perform measurements at temperatures as low as 50 K. 

## 3. Mathematical Models

The TR signal can be written as follows with the notation for the multilayer coefficients [24]:(4)$$\theta_{TR}=\frac{-Q}{2\pi }\;\int_{0}^{\infty }\frac{D}{C}e^{-\frac{{\left(\lambda d\right)}^{2}}{8}}\lambda \;d\lambda$$
where *λ* = Hankel variable, *Q* = input pump beam intensity, d the beam spot, and C and D are coefficients, defined as:(5)$$\left[\begin{array}{cc} An & Bn\\ Cn & Dn \end{array}\right]=M_{3}{\;M}_{32}{\;M}_{2}{\;M}_{21}{\;M}_{1}$$
where
(6)$$A_{n}=cos\;h\;\left(\sigma_{n}l_{n}\right)=D_{n}\;;B_{n}=\frac{sinh(\sigma_{n}l_{n})}{k_{⊥n}\sigma_{n}};C_{n}=-k_{⊥n}\sigma_{n}sinh\;h\;(\sigma_{n}l_{n})$$
where $k_{∥}$ = in-plane thermal conductivity, and $k_{⊥}$ = cross-plane thermal conductivity.

The thermal boundary resistance at the interface is defined as follows:(7)$$M_{n,n+1}=\left[\begin{array}{cc} 1 & -R_{n,n+1}\\ 0 & 1 \end{array}\right]$$
for the continuity condition at the interface,
ϕ=ampltiude,
θ=phase of the thermal response
(8)$${\phi (l}_{+})={\phi (l}_{-})=\frac{\theta \left(l_{-}\right)-\theta \left(l_{+}\right)}{R}$$
and
(9)$$\sigma^{2}=\frac{k_{∥}}{k_{⊥}}\left(\lambda^{2}+i\omega \rho C\right)\;for\;TR$$

*ω =* pump modulation frequency, ρC = volumetric specific heat, and $\frac{k_{∥}}{k_{⊥}}=\eta$ is the anisotropy index. In order to obtain the expression of the one dimensional PTR signal, we can calculate
(10)$$\theta_{PTR}=-\frac{D}{C}$$
and
(11)$$\sigma^{2}=\frac{1}{k_{⊥}}\left(i\omega \rho C\right)\;for\;PTR$$

Figure 4 and Figure 5 show the sensitivity vectors drawn in relation to each other to show that not all parameters can be determined from FDTR data. Figure 5b shows the sensitivity vector S(kr) vs. S(Ceff) (volumetric heat capacity), eliminating the uncertainty involved in measuring both cross-plane and in-plane thermal properties as the anisotropy has been kept fixed. As can be seen, the two parameters are linearly related, so they cannot be determined simultaneously from the experiment. In the rest of the article, we propose two scenarios to circumvent this problem.

## 4. Accuracy and Precision of PTR and TR Methods

Each measurement method is characterized by accuracy and precision. Precision is relatively easy to define because it is defined as a measure of standard deviation (standard precision) or as two measured deviations (extended precision). Ultra-precise estimations of thermal parameters (thermal conductivity, boundary resistance) using frequency domain thermoreflectance (FDTR) are detailed in Table 1 with GaAs as the substrates for both in and cross-plane symmetry using a machine learning approach. The value of heat capacity was assumed to be 0.325 J/g°C from PTR experiments. The isotropic index of the GaAs is rightfully 1, as the ratio matches Table 1. A detailed discussion of these results is provided in Section 5.6. 

In our case, we tested the accuracy of both methods using a 2 µm sample of AlGaAs superlattice. The following analysis represents the obtained experimental results—amplitude and phase. PTR and TR data were validated for the existing literature and measured using each of these methods. For instance, the TR and PTR results for the AlGaAs sample are presented in Figure 6, with (a) showing the TR phase and (b) the PTR amplitude and phase, respectively.

The obtained results are in agreement with the predicted value given in the literature and measured using PTR. The result was 12 W/mK [27] and agreed very well with the reported data [28].

## 5. Application of Methods

### 5.1. Application of PTR Method for Characterization of Infrared Absorption Coefficient of Thin Film

The samples of interest were films of Al_0.33_Ga_0.67_As with and without C doping grown on a Zn-doped GaAs substrate using molecular beam epitaxy (III-V-MBE) [26]. In fact, to study nanomaterials, one can easily modify the above formula to add other layers (the so-called two layers model: nanolayer–substrate):(12)$$S\left(f,\beta_{IR}\right)\propto \int_{0}^{L_{1}}\beta_{IR1}\cdot {∆T}_{1}\left(z,f\right)\cdot exp\left(-\beta_{IR1}\cdot z\right)dz+\int_{L_{1}}^{L_{2}}\beta_{IR2}\cdot {∆T}_{2}\left(z,f\right)\cdot exp\left(-\beta_{IR2}\cdot z\right)dz$$

The formulas for ${∆T}_{1}$ and ${∆T}_{2}$ can be found in Ref. [26]. Figure 7 shows the IR signal received for undoped and C-doped AlGaAs thin layers.

The signals were normalized to the substrate sample, resulting in amplitude ratios of 1 and a phase difference of 0°. At lower frequencies, the signal was predominantly from the substrate, while at higher frequencies, it was associated with the thin film. To analyze thinner samples in the nanometer range, higher frequencies must be utilized. The thermal properties (κ and D) of the samples are almost the same, so the signals differ due to the change in *β_IR_* and thermal boundary resistance (*R_B_*). For the undoped sample, *β_IR_* = 0 and *R_B_ =* ~10^−9^ m^2^K/W, and for the doped AlGaAs, one can find values of *β_IR_*. Figure 8 shows the *β_IR_* versus the inverse of wavelength to follow a well-known dependence.
(13)$$\beta_{IR}\left(\lambda \right)=a\left(N\right)\sqrt{\frac{1}{\lambda }}\left(e^{\frac{-b}{\lambda }}-e^{\frac{-c}{\lambda }}\right)$$
where a, b, and c are coefficients. Figure 8 shows the estimated *β_IR_* vs. 1/wavelength. When compared to the undoped sample, the doped sample also has a different *R_B_*, which is in the order of 10^−7^ m^2^K/W due to modulation doping. It is worth mentioning that it was not possible to characterize the doped sample using FTIR spectroscopy, as well as Hall or capacitance–voltage spectroscopy.

### 5.2. Thermal Properties of III-V Semiconductor Superlattices at Room Temperature

The AlAs/GaAs superlattices were prepared on GaAs substrate wafers using molecular beam epitaxy (MBE). The initial sample comprises a 150 nm GaAs buffer layer and a superlattice comprising 10 iterations of approximately 26 nm AlAs and 26 nm GaAs, concluding with a GaAs layer. The second sample involves a 150 nm GaAs buffer layer and a superlattice, featuring 100 repetitions of nominally 2.6 nm GaAs and 2.6 nm AlAs, with an additional 5 nm GaAs layer on the top. Subsequently, a Ti layer with a thickness of 50 nm was deposited on both of these samples [24,28].

One of the examined superlattices of AlAs/GaAs consists of 20 layers, whereas the second one consists of 200 layers, exhibiting equivalent thermal resistance (thermal resistances of the individual layers + $R_{B}$ between individual layers) [27] that can be written as follows:(14)$$R_{eq}=N\left(2R_{B}+\frac{l_{GaAs}}{k_{GaAs}}+\frac{l_{AlAs}}{k_{AlAs}}\right)=\frac{L}{k_{eq}}$$
whereas the equivalent thermal diffusivity has the form:(15)$$D_{eq}=\frac{k_{eq}}{{\left(\rho c\right)}_{eq}}=\frac{l_{GaAs}}{l}{\left(\rho c\right)}_{GaAs}+\frac{l_{AlAs}}{l}{\left(\rho c\right)}_{AlAs}$$
where *N*—number of superlattice periods; *k_GaAs_*/*k_AlAs_*—κ of GaAs/(AlAs) layer; *l_GaAs_*/*l_AlAs_*—thickness of the GaAs/(AlAs) layer; *L*—total thickness of the superlattice; *k_eq_*—equivalent κ of the superlattice layer; *l*—period of the superlattice; *c_eq_*/*c_GaAs_*/*c_AlAs_*—equivalent volumetric heat capacity and volumetric heat capacity of GaAs or AlAs material, respectively; and *ρ_eq_*—equivalent density of GaAs (*ρ_GaAs_*) or AlAs (*ρ_AlAs_*) material. The behavior of the IR signal phase and amplitude with ER modulation frequency for both types of superlattice samples is presented in Figure 9, whereas the computed thermal parameters are shown in Table 2. The thermal properties of a superlattice are observed to be contingent on its structure, particularly in the ratio of the period thickness to the total thickness of the superlattice.

As one can see, all the values are within the experimental errors. In fact, it is worth looking closer at the thermal boundary resistance for 100× 2.6 nm AlAs and 2.6 nm GaAs. In this next discussion, we move to challenges.

### 5.3. Temperature-Dependent Measurements of Superlattice Samples

The advantage of IR over other methods described in this paper is that temperature-dependent measurements can be easily performed. The modification of the system shown in Figure 7 is to place the Linkam cell in the focal length of one of the off-axial mirrors. The IR signal obtained for different temperatures can be found in Figure 10. 

### 5.4. Thermal Characterization of Polymer (PPP) Samples

In this study, the determination of the thermal properties of push–pull azo-functionalized side chain polymer thin films (Figure 11) was performed. The samples were covered by a metallic transducer. The experimental setup depicted in Figure 1 was employed to record the phase and amplitude of the IR signal and is shown in Figure 12.

The values of κ, effectivities (ε), D, and volumetric heat computed are presented in Table 3.

It was found that in order to enhance heat transport in the side-chain azo polymer, elongating the side-chain structure through the multiplication of azo molecules and increasing the content of the push–pull molecule type are both essential (Figure 11) [29].

### 5.5. Application of TR Phase to Characterize Material

The analyzed sample was a 52 nm AlAs/GaAs superlattice of periodicity prepared on a GaAs substrate. The phase of the collected TR signal superlattice collected from the sample for three different temperatures of −50, 0, and 50 °C is presented in Figure 13 [24].

The signals were analyzed using the model presented in Section 3 and the in-plane κ values were obtained. It is worth noting that infrared radiometry was used to determine cross-plane parameters while thermo-reflection was used for the in-plane parameters. It is essential to mention that determining the in-plane parameters of the layer requires knowledge of its specific heat capacity. Figure 14 shows the cross-plane and in-plane κ values of the AlAs/GaAs superlattice [24] computed as a function of temperature. It can be seen that the thermal anisotropy of the sample is expressed as a two times higher value of in-plane thermal conductivities compared to the through-plane values. This finding aligns with the phenomenon of phonon reflections occurring at the interfaces of GaAs and AlAs.

### 5.6. Measurements of Both Amplitude and Phase for Better Accuracy and Precision

Deep learning methods are well-suited for analyzing experimental data, with FDTR showing a significantly lower standard error compared to time domain analysis. In this context, weight vectors are employed to fine-tune the thermal parameters and achieve the global minimum. Next, we examine the impact of the deep learning model on error propagation assessment. The deep-learning Levenberg–Marquardt (LM) model offers a notable advantage in accurately evaluating uncertainty by effectively converging on local minima. As a result, the predictions generated by the deep learning model are superior to those obtained through the trust region (TR) method. Although both strategies are commonly used for non-linear least squares fitting, the back-propagation (AP) model outperforms the previous results in both time and frequency domains. This supports the notion that in linear least squares fitting, a more precise initial prediction consistently leads to more accurate error estimations [26].

The chi-square test is utilized to determine the statistical significance of the relationship between two categorical variables, specifically amplitude and phase when used as input parameters. Regarding FDTR, it has relatively low sensitivity in the frequency range of 10 kHz to 100 kHz, making this range the optimal starting point for maximum likelihood estimation. A narrower range increases the likelihood of convergence to or near the global minimum, even with an imperfect initial prediction when the model operates within the ideal sensitivity range. Based on the data in Table 1, it can be concluded that incorporating an additional information channel, specifically the amplitude of the thermoreflectance signal, reduces error estimation. Figure 15 presents the experimental data.

The obtained value of thermal conductivity of 54 W/mK is in very good agreement with the literature values. The value of 55 W/mK for GaAs was reported in Ref. [27]. The advantage of the described methods over TDTR is that it is possible to measure the thermal diffusivity of the samples, which is one of the most important parameters used in modeling heat propagation in the sample.

### 5.7. Superlattice

To verify that observing the TR amplitude improves precision, we investigated the superlattice described in Section 5.5. The experimental data are presented in Figure 16. 

The results obtained for in-plane thermal conductivity, cross-plane thermal conductivity and thermal boundary resistance ***R_th_*** are presented in Table 4 As in previous analysis, incorporating amplitude data improves measurement precision to approximately 3–4 %, compared to around 10% when using only phase-based approach.

### 5.8. Application of Thermoreflectance for PEDOT:PSS

The estimation of the thermal transport properties of the poly(3,4-ethylenedioxythiophene): poly(styrenesulfonate) (PEDOT:PSS) sample with comparable transport conductivity has been validated by Ref. [8]. The in-plane thermal transport properties are unknown because it is impossible to measure all the parameters simultaneously using the single method and with similar distribution over the scanning variables. Hence, the most important way to tackle this issue is to assume that the anisotropy index is kept in a fixed state before the start of the final parameter estimation. For any known anisotropy, the estimation of the thermal parameters is possible. Figure 17a shows the sensitivity analysis done for PEDOT:PSS by using signal parameters set as constant in Table 5. Furthermore, the reduced chi-squared statistics in Figure 17b also suggest that for the optimized anisotropy, the measurements can be optimized for ultraprecise measurements. Before measurement, the sample was prepared as follows: A layer of aluminum and then gold was applied to Pedot:pSS. Figure 18 shows the experimental results obtained for PEDOT:PSS, where (a) illustrates the normalized amplitude (R) and (b) depicts the phase (in degrees) as a function of pump modulation frequency variation. Here, we aim to characterize the sample using an alternative approach. To achieve this, the anisotropic index (η) was optimized based on the reduced chi-square, as shown in Table 6. Subsequently, the in-plane thermal conductivity was calculated using the values of η and the cross-plane thermal conductivity provided in the same table.

The determination of the anisotropic index for the samples has an elemental relation with a reduced chi-square. Optimized η values have been tabulated with the best-fitting results for the cross-plane analysis. The correlation results between the thermal parameters are in good accordance with the previous work [8]. The cross-plane values estimated are a good analogy for the optimized anisotropy (~3.67) for the given sample. Table 7 shows obtained results for the optimized in-plane and cross-plane thermal transport properties using Formula (4).

## 6. Conclusions

In this article, we present a unique strategy for measuring the thermal properties of inorganic and organic thin films and superlattice samples. Theoretically, there is no limit to the thickness of a thin film, but then, a thicker transducer (metal) layer must be used. The presented strategy also self-checks whether the obtained values are unique by measuring cross-plane thermal conductivity and thermal boundary resistance with two methods. Thanks to the frequency measurement methods, the signal also depends on the volumetric heat capacity. This study significantly advances the understanding of thermal transport properties in thin films through the application of photothermal radiometry (PTR) and thermoreflectance (TR) techniques. These methods have demonstrated high accuracy in measuring thermal conductivity, diffusivity, and boundary resistance, particularly in multilayered materials exhibiting thermal anisotropy. The PTR method’s ability to perform temperature-dependent measurements and the TR method’s sensitivity to in-plane thermal transport highlight their efficacy and versatility. Applications to both organic and inorganic thin films, including AlGaAs layers and III-V semiconductor superlattices, have confirmed the methods’ reliability and precision. The findings reveal that thermal properties are highly dependent on superlattice structure and polymer side-chain configurations, providing valuable insights for designing materials with improved thermal characteristics. These advancements pave the way for further exploration of complex material systems and the refinement of measurement techniques, contributing significantly to materials science and engineering.

## Figures and Tables

**Figure 1 nanomaterials-14-01711-f001:**
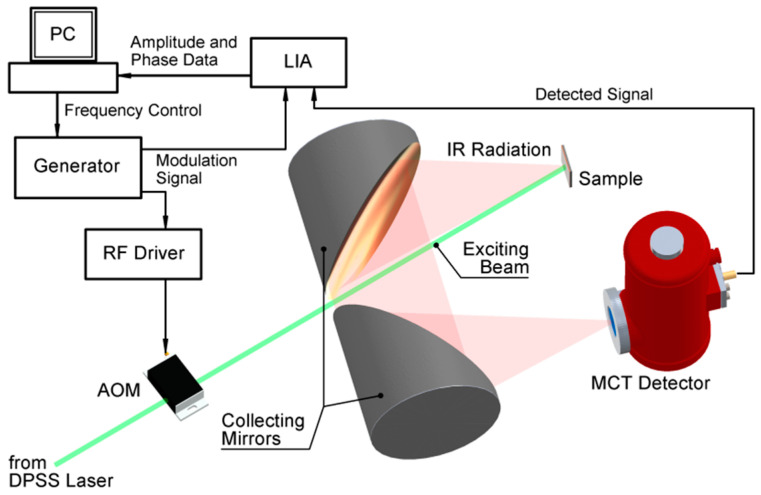
The PTR experimental set-up used in the study.

**Figure 2 nanomaterials-14-01711-f002:**
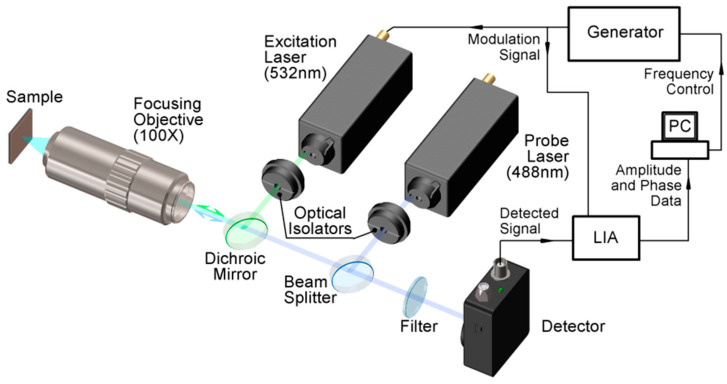
Typical FDTR experimental set-up.

**Figure 3 nanomaterials-14-01711-f003:**
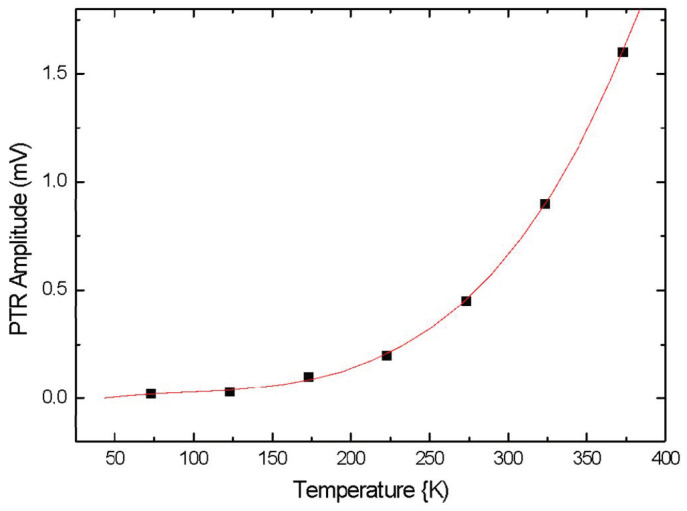
PTR signal dependence vs. temperature for GaAs wafer cover by 50 nm titanium layer at 1 kHz. Fitting confirms *T*^3^ law.

**Figure 4 nanomaterials-14-01711-f004:**
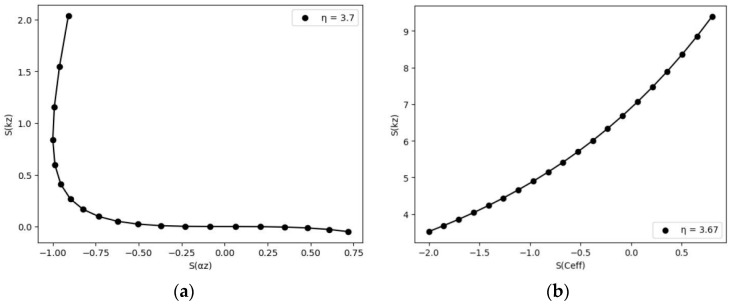
Sensitivity coefficients. (**a**) Cross-plane thermal conductivity vs. cross-plane diffusivity for different anisotropy coefficients; (**b**) volumetric heat capacity vs. cross-plane thermal conductivity. For sensitivity analysis for PEDOT:PSS and parameters depicted in Table 5.

**Figure 5 nanomaterials-14-01711-f005:**
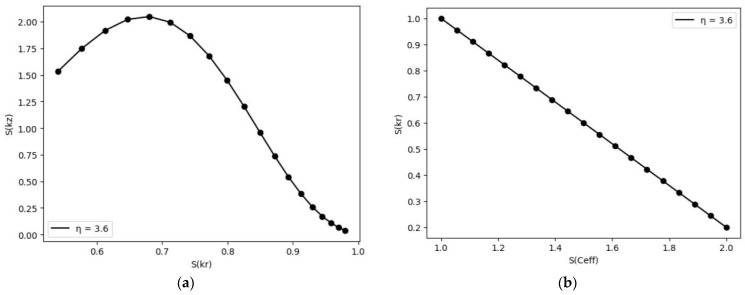
Sensitivity coefficient. (**a**) Cross-plane thermal conductivity vs. in-plane thermal conductivity for optimized anisotropy coefficients; (**b**) volumetric heat capacity vs. in-plane thermal conductivity.

**Figure 6 nanomaterials-14-01711-f006:**
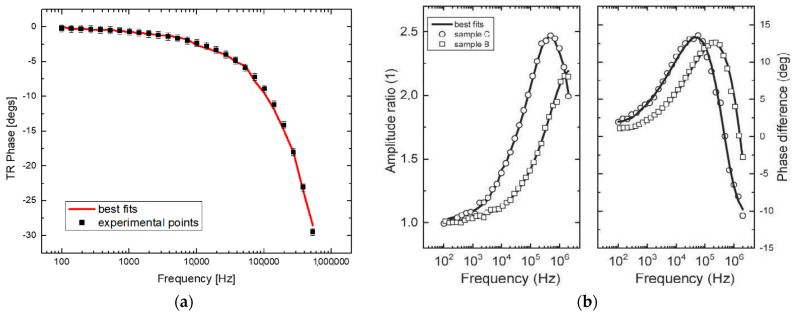
(**a**) Measured TR phase of 2 μm AlGaAs thin layer covered by 100 nm of gold between 100 Hz and 500 kHz [24]. (**b**) PTR results of these samples (covered by 50 nm Ti not Au transducer), sample C is AlGaAs with 2 μm thickness and sample D has 512 nm thickness [27].

**Figure 7 nanomaterials-14-01711-f007:**
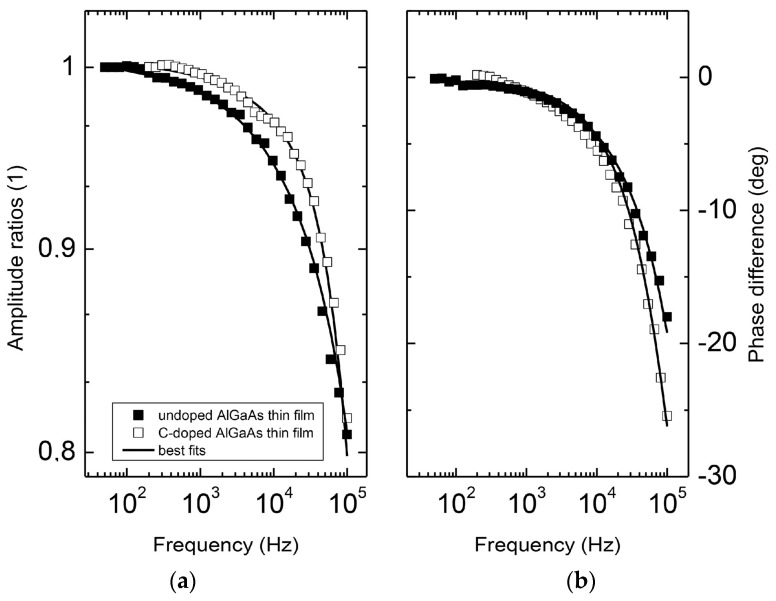
Variation of (**a**) amplitude ratios and (**b**) phase difference with frequency for Al_0.33_Ga_0.67_As alloys with and without C doping [25].

**Figure 8 nanomaterials-14-01711-f008:**
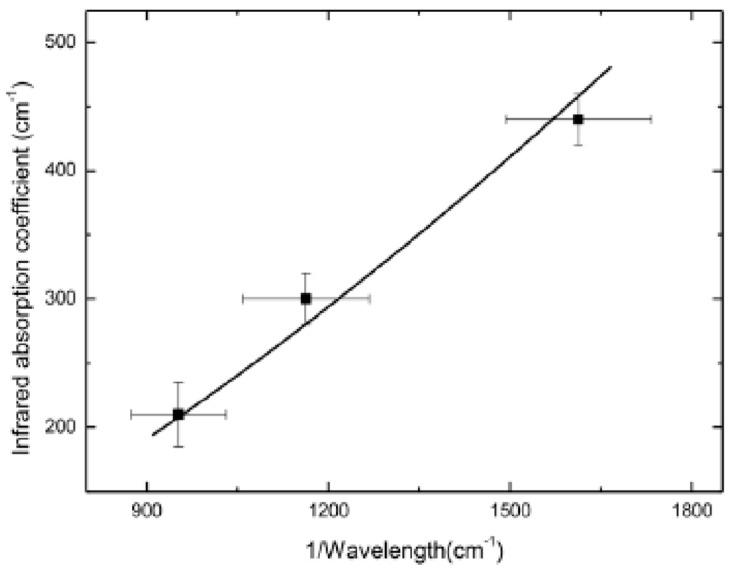
*β_IR_* of C-doped Al_0.33_Ga_0.67_As thin film (point) as a function of inverse *λ*. Solid line is *β_IR_* calculated using Equation (10) [25].

**Figure 9 nanomaterials-14-01711-f009:**
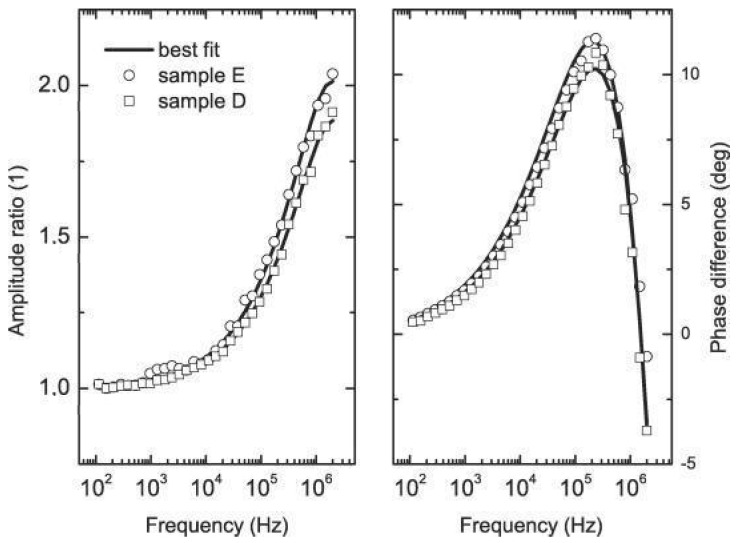
Variation of amplitude ratios and phase difference with frequency for AlAS/GaAs superlattice samples: sample D (10× 26 nm AlAs and 26 nm GaAs) and sample E (100× 2.6 nm AlAs and 2.6 nm GaAs) [27].

**Figure 10 nanomaterials-14-01711-f010:**
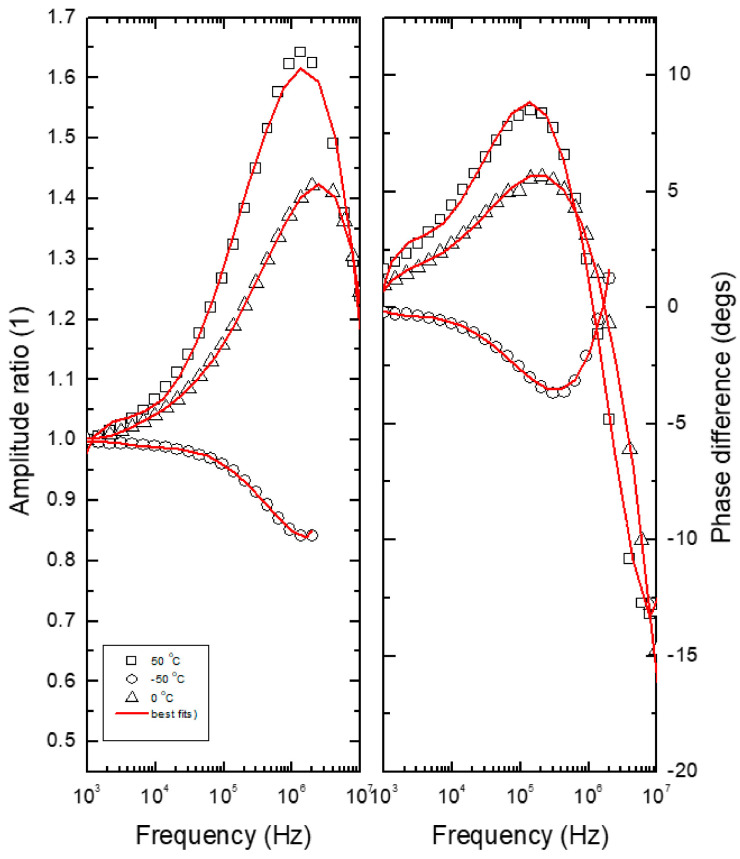
Variation of amplitude ratios and phase difference with frequency for a sample at three different temperatures: −50 °C, 0 °C, and 50 °C [24].

**Figure 11 nanomaterials-14-01711-f011:**
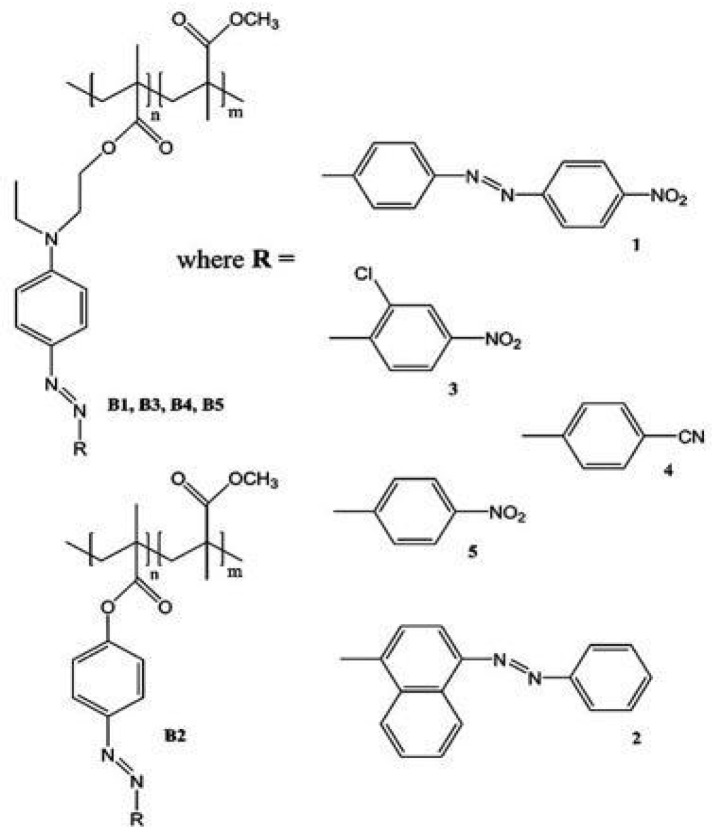
Molecular structures of push–pull type (B1, B3, B4, and B5) and azobenzene-type (B2) polymers [29].

**Figure 12 nanomaterials-14-01711-f012:**
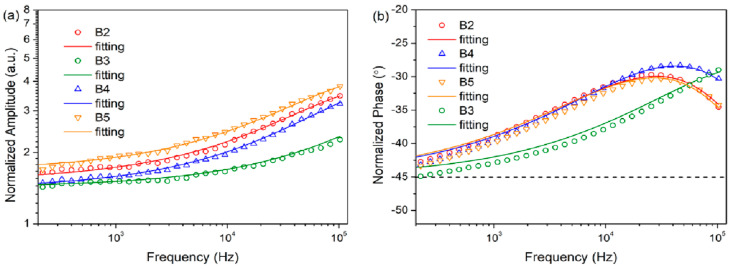
(**a**) Variation of normalized amplitude and (**b**) normalized phase with modulation frequency for the B2–B5 azo polymer thin films in their trans state [29].

**Figure 13 nanomaterials-14-01711-f013:**
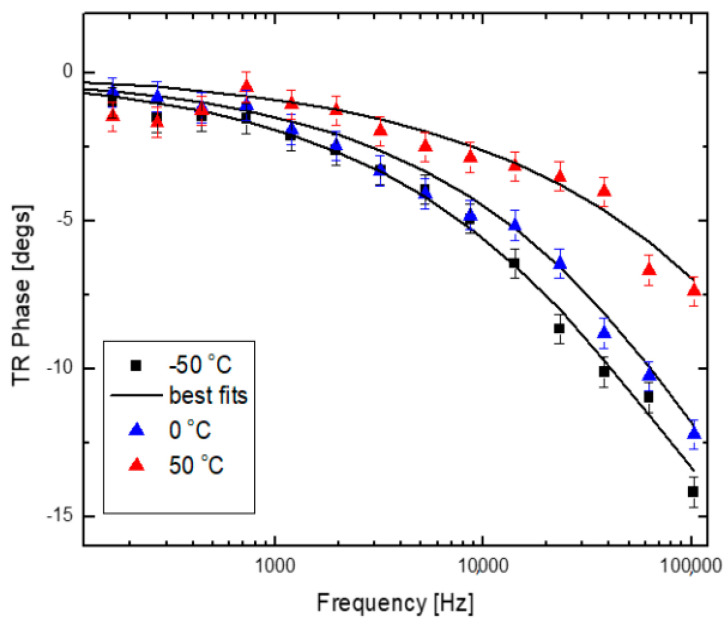
FDTR measurement of a AlAs/GaAs superlattice sample at different temperatures [24].

**Figure 14 nanomaterials-14-01711-f014:**
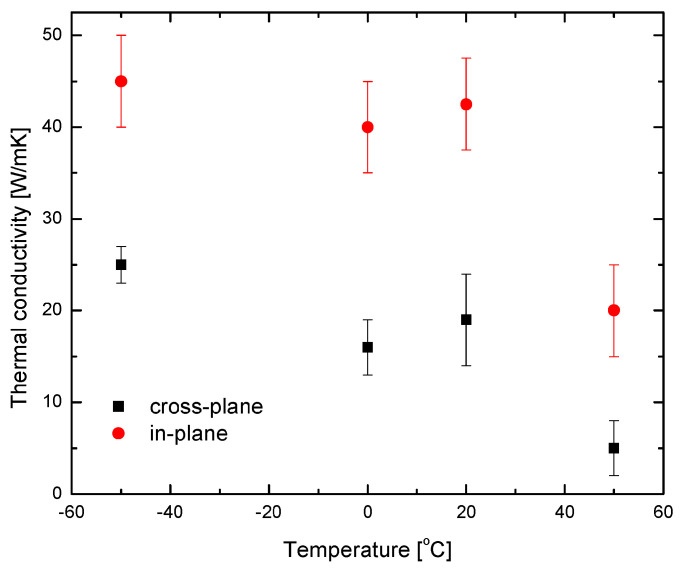
Variation of in-plane and cross-plane thermal conductivity of the AlAs/GaAs superlattice with temperature [24].

**Figure 15 nanomaterials-14-01711-f015:**
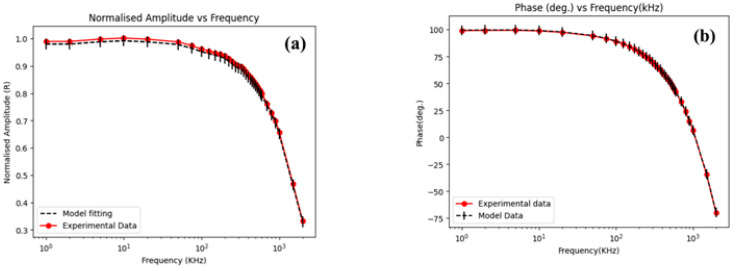
(**a**) Normalized amplitude (R); (**b**) phase (deg.) vs. modulation frequency (kHz) [25].

**Figure 16 nanomaterials-14-01711-f016:**
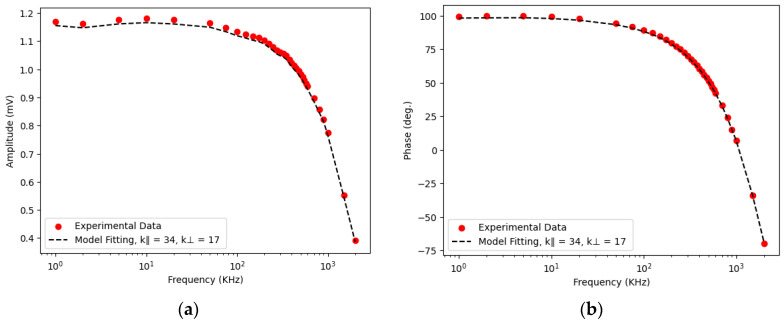
(**a**) Normalized amplitude (mV) and (**b**) phase (degrees) as a function of pump modulation frequency variation.

**Figure 17 nanomaterials-14-01711-f017:**
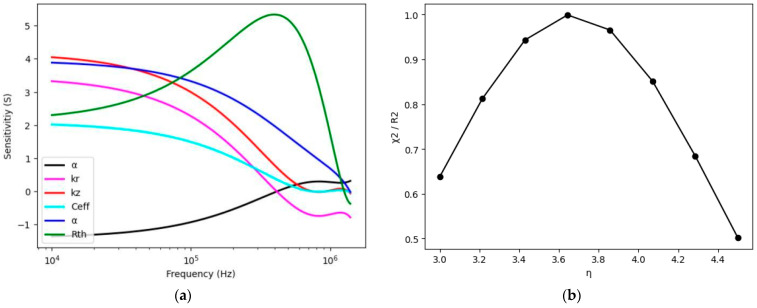
(**a**) Sensitivity analysis of thermal properties of PEDOT:PSS (w.r.t.) scanning modulation frequency (Hz) for parameters described in Table 5; (**b**) Reduced chi-square statistic vs. anisotropic coefficient (η).

**Figure 18 nanomaterials-14-01711-f018:**
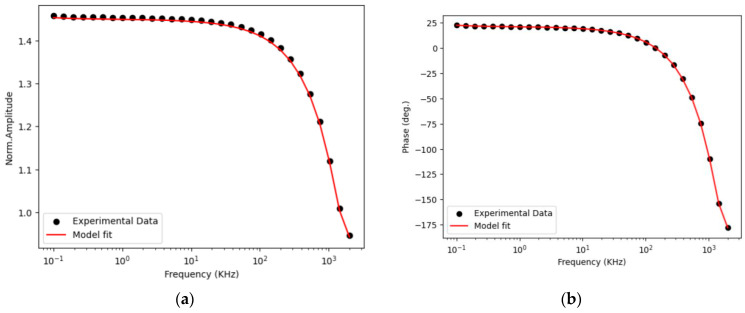
(**a**) Normalized amplitude (R); (**b**) phase (degree.) vs. variation in pump modulation frequency for PEDOT:PSS.

**Table 1 nanomaterials-14-01711-t001:** Error statistics (MAPE) of thermal parameters [26].

Geometrical Symmetry	k_GaAs_ (W/m-K)	α_Au/GaAs_ (×10^−5^ m^2^/s)	Thermal Boundary Resistance (m^2^K/W)
Cross-plane	53.35 ± 1.23	(0.989 ± 0.145)	7.7928 × 10^−7^ ± 4.8211%
In-plane	53.3125 ± 0.12 (isotropic)	-	7.7928 × 10^−7^ ± 4.2911%

**Table 2 nanomaterials-14-01711-t002:** Thermal parameters of AlAs/GaAs superlattice at room temperature.

	10 × 26 nm AlAs and 26 nm GaAs	100 × 2.6 nm AlAs and 2.6 nm GaAs
*k_eq_* (W/m K) ^1^	19.0 ± 5.0	10.0 ± 3.0
*k_eq_* (W/m K) ^2^	15	7.5
*D_eq_* (m^2^/s) ^1^	(1.0 ± 0.2) × 10^−5^	(5.0 ± 1.3) × 10^−6^
*D_eq_* (m^2^/s) ^3^	(1.1 ± 0.3) × 10^−5^	(6.1 ± 2.4) × 10^−6^
*R_eq_* (m^2^ KW^−1^) ^1^	(3.0 ± 0.8) × 10^−8^	(3.0 ± 0.6) × 10^−8^
*R_eq_* (m^2^ KW^−1^) ^4^	(2.0 ± 0.9) × 10^−8^	(4.4 ± 2.1) × 10^−8^
*R_eq_* (m^2^ KW^−1^) ^2^	~2 ×10^−8^	~2 ×10^−7^

^1,2^ from Ref. [27], ^3^ calculated using Equation (14), ^4^ and using Equation (15).

**Table 3 nanomaterials-14-01711-t003:** Thermal parameters of the samples B2–B5 [8].

Sample	α 10^−7^ m^2^s^−1^	ε Ws^−0.5^ m^2^K^−1^	κ Wm^−1^K^−1^	C_v_ 10^6^ Jm^−3^K^−1^
B2	1.45	573	0.22	1.50
B3	1.75	450	0.19	1.07
B4	1.48	509	0.20	1.32
B5	1.01	580	0.18	1.82

**Table 4 nanomaterials-14-01711-t004:** Error statistics (MAPE) in the thermal model.

Approach	Geometrical Symmetry	Sample	*k_AlAs/GaAs_* (W/m-K)	α_AlAs/GaAs_ (×10^−5^ m^2^/s)	*R_th_* (m^2^W/K)
Frequency Scan	cross-plane	GaAs/AlAs 52 nm	~17.425 ± 0.235	~1.1023 ± 0.0252	~(6.127 ± 0.120) × 10^−9^
Frequency Scan	in-plane	GaAs/AlAs	~36.154 ± 1.32		

**Table 5 nanomaterials-14-01711-t005:** Parameters set as a constant for FDTR.

Parameters	Assumed Values
Spot Size (µm)	2.42 (FDTR)
*κ* (glass) W/(m·K)	1.38
k_1_ = k_Au_ (W/m-K)	105 ± 9.9
α_1_ = α_Au_ (m^2^/s)	1.23 × 10^−5^
d_1_ = d_Au_ (nm)	20
d_2_ (nm) (PEDOT:PSS)	480
d_Al_ (nm)	50
d_Glass_ (mm)	1
k (Al) (W m^−1^ K^−1^)	200
α_1_ = α_Glass_ (m^2^/s)	0.34 × 10^−6^
*ρ*C_v_ (Al) (J·cm^−3^·K^−1^)	0.98 × 10^6^
*ρ*C_v_ (Au) (J·m^−3^·K^−1^)	3.6 × 10^6^

**Table 6 nanomaterials-14-01711-t006:** Anisotropy index (η) values for different fittings with given R^2^.

η	χ^2^/R^2^
3	0.88
3.3	0.9
3.45	0.92
3.6	0.9367
3.67	0.9645
3.8	0.917
4.11	0.89
4.5	0.88

**Table 7 nanomaterials-14-01711-t007:** Table for the optimized in-plane and cross-plane thermal transport properties in the PEDOT:PSS estimated using Formula (4).

Geometry	In-Plane Thermal Conductivity (W/mK)	Cross-Plane Thermal Conductivity (W/m-K)	Thermal Diffusivity (×10^−7^ m^2^/s)	Thermal Boundary Resistance (×10^−9^ m^2^ KW^−1^)	ꭕ^2^
	0.67	0.182	1.48254	1.6245	94.914

## Data Availability

The raw data supporting the conclusions of this article will be made available by the authors on request.

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
