# Peer review of "Photothermal Infrared Radiometry and Thermoreflectance—Unique Strategy for Thermal Transport Characterization of Nanolayers"

_nanomaterials, 2024, doi:10.3390/nano14211711_

Round 1

Reviewer 1 Report

Comments and Suggestions for Authors

Photothermal infrared radiometry and thermoreflectance – unique strategy for thermal transport characterization of nanolayers

Good and interesting manuscript and research.

In your introduction section, I recommend you to stablish what is the value added of your work, compared to previously published investigations. Why it is attractive to readers?

The setup you describe in Section 2 follows any ASTM standard? or what is the reference for this setup?

Figure 3 needs work to do. It seems that during the PDF conversion it was distorted.

Figure 5 could be improved, it looks crisp, the letters in the box are blurry.

In your conclusions, what is the "punching line"? why your research is important or attractive to readers? what did you found?

In the references, I highly recommend the authors to search and apply more recent information, at least >2022

Comments on the Quality of English Language

English grammar could be improved to give more flow to the reading and make it easier to the readers to follow 

Author Response

omment - Good and interesting manuscript and research.

  1. In your introduction section, I recommend you to establish what is the value added of your work, compared to previously published investigations. Why it is attractive to readers?

Ans - The introduction section and the literature review have been updated for the same as suggested. Kindly check the section in “Introduction” starting from Line 38-39 onwards. In this paper we discuss main problem of measuring all thermal transport parameters. They cannot be estimated simultaneously. We proposing two approach : one measuring volumetric heat capacity and the second find the optimum anisotropy index ( ? =k∥/k⊥= ratio of in-plane to cross-plane thermal conductivity) from the fittings. Updated info in paper marked in red.

  1. The setup you describe in Section 2 follows any ASTM standard? or what is the reference for this setup?

Ans - Yes

  1. Figure 3 needs work to do. It seems that during the PDF conversion it was distorted, Figure 5 could be improved, it looks crisp, the letters in the box are blurry.

Ans- Fig 3 and Fig 5 has been changed due to some changes in the outline of the article. Kindly suggest if the adjoining graphs need more updates.

  1. In your conclusions, what is the "punching line"? why your research is important or attractive to readers? what did you found?

Ans- The line in the conclusion section line 595 and onwards shows the elementary underlying statement of the research and the exclusiveness of the research.a unique strategy for measuring the thermal properties of inorganic and organic thin films and superlattice samples. Check also answer to question 1

Reviewer 2 Report

Comments and Suggestions for Authors

The authors studied the measurement technique of thermal conductivity. The author proposed the technique using both photothermal radiometry (PTR) and thermoreflectance (TR). Using both techniques, it is expected to combine the merits of both techniques. In addition, the authors measured the thermal conductivities of AlAs/GaAs superlattice and PMMA film, showing the reliability of this technique. However, there are some concerns about the introduction and the discussion sections. If the authors appropriately revise the manuscript, this study will meet the criteria for the publication in nanomaterials.

Comment 1: The authors measured the thermal conductivities of AlAs/GaAs superlattice as a function of temperature. Does the thermal conductivity measured by PTR and TR technique agree with the ones measured by other conventional techniques? If the authors cannot implement some conventional techniques for comparison, the authors should compare the authors’ data with the previously reported results at least.

Comment 2: In Figure 15, the authors found that the effective thermal conductivity of PMMA film in Au/Ni/PMMA structure was higher than that of PMMA film in Au/PMMA structure.

Used model in the phase fitting for measurement is changed when the Ni is inserted.

Is this considered?

If it is considered, how did the authors remove the contribution of Ni film in Au/Ni/PMMA structure? I wonder if the effective thermal conductivity of PMMA film in Au/Ni/PMMA includes the contribution of Ni film.

Comment 3: In Figure 15, the authors measured the thickness dependence of the effective thermal conductivity of PMMA film. Then, the authors showed the effective thermal conductivity of PMMA film with the thickness of 1 nm. Is it possible to form uniform film with 1 nm? What is the fabrication process?

When it is possible to form 1nm film, is it possible to measure the effective thermal conductivity of the PMMA film with the thickness of 1nm? I wonder if PMMA is really formed as the film structure because the thickness is too thin.

Comment 5: The authors described 3w method, TDTR, and FDTR in the introduction section. The 2w method is also one of the thermal conductivity measurement methods. The authors cannot ignore this method. Please describe 2w method and cite the relatively-recent related research.

Comment 6: In the introduction, the authors did not mention the thermal management application although they mentioned that studying thermal physical parameters is important. There are various thermal management applications: thermal switch (Nano Lett. 22, 6105 (2022).), thermoelectric generator (Nano Lett. 22, 4131 (2022).), thermal diode, etc. The authors should comment on the thermal management application in the beginning of introduction section.

Author Response

The authors studied the measurement technique of thermal conductivity. The author proposed the technique using both photothermal radiometry (PTR) and thermoreflectance (TR). Using both techniques, it is expected to combine the merits of both techniques. In addition, the authors measured the thermal conductivities of AlAs/GaAs superlattice and PMMA film, showing the reliability of this technique. However, there are some concerns about the introduction and the discussion sections. If the authors appropriately revise the manuscript, this study will meet the criteria for the publication in nanomaterials.

Ans - the introductionand the discussions have been altered as been discussed in the last comment section. Kindly suggest any further comments on the same. The conlcuions have been rewritten.

We replaced PMMA section with our results of PEDOT:PSS. Please check application of thermoreflectance to PEDOT:PSS section.

Comment 1: The authors measured the thermal conductivities of AlAs/GaAs superlattice as a function of temperature. Does the thermal conductivity measured by PTR and TR technique agree with the ones measured by other conventional techniques? If the authors cannot implement some conventional techniques for comparison, the authors should compare the authors’ data with the previously reported results at least.

Comment 2: In Figure 15, the authors found that the effective thermal conductivity of PMMA film in Au/Ni/PMMA structure was higher than that of PMMA film in Au/PMMA structure.

Ans - We replaced PMMA section with our results of PEDOT:PSS. Please check application of thermoreflectance to PEDOT:PSS section.

Is this considered?

Ans - We replaced PMMA section with our results of PEDOT:PSS. Please check application of thermoreflectance to PEDOT:PSS section.

Comment 3: In Figure 15, the authors measured the thickness dependence of the effective thermal conductivity of PMMA film. Then, the authors showed the effective thermal conductivity of PMMA film with the thickness of 1 nm. Is it possible to form uniform film with 1 nm? What is the fabrication process?

When it is possible to form 1nm film, is it possible to measure the effective thermal conductivity of the PMMA film with the thickness of 1nm? I wonder if PMMA is really formed as the film structure because the thickness is too thin.

Ans - We replaced PMMA section with our results of PEDOT:PSS. Please check application of thermoreflectance to PEDOT:PSS section.

Comment 5: The authors described 3w method, TDTR, and FDTR in the introduction section. The 2w method is also one of the thermal conductivity measurement methods. The authors cannot ignore this method. Please describe 2w method and cite the relatively-recent related research.

Ans. Done

Comment 6: In the introduction, the authors did not mention the thermal management application although they mentioned that studying thermal physical parameters is important. There are various thermal management applications: thermal switch (Nano Lett. 22, 6105 (2022).), 

thermoelectric generator (Nano Lett. 22, 4131 (2022).), thermal diode, etc. The authors should comment on the thermal management application in the beginning of introduction section.

Ans - Done

Round 2

Reviewer 2 Report

Comments and Suggestions for Authors

The manuscript is properly revised. However, there is a minor mistake. The name of the first author is wrong in Ref.1.

If it is revised, this meets the criteria for the publication.

Author Response

The suggested changes have been rectified. Kindly check the same.